# Breast Cancer Detection in Thermographic Images via Diffusion-Based Augmentation and Nonlinear Feature Fusion

Sepehr Salem, M. Moein Esfahani, Jingyu Liu, Vince Calhoun

*Computer Science Department*

*Center for Translational Research in Neuroimaging and Data Science (TReNDS), GSU, GA Tech, and Emory*

Atlanta, GA, USA

{ssalemghahfarokhi1, mesfhani1, jliu75, vcalhoun}@gsu.edu

*Abstract*—Data scarcity hinders deep learning for medical imaging. We propose a framework for breast cancer classification in thermograms that addresses this using a Diffusion Probabilistic Model (DPM) for data augmentation. Our DPM-based augmentation is shown to be superior to both traditional methods and a ProGAN baseline. The framework fuses deep features from a pre-trained ResNet-50 with handcrafted nonlinear features (e.g., Fractal Dimension) derived from U-Net segmented tumors. An XGBoost classifier trained on these fused features achieves 98.0% accuracy and 98.1% sensitivity. Ablation studies and statistical tests confirm that both the DPM augmentation and the nonlinear feature fusion are critical, statistically significant components of this success. This work validates the synergy between advanced generative models and interpretable features for creating highly accurate medical diagnostic tools.

*Index Terms*—breast cancer, thermography, diffusion probabilistic model, deep learning, data augmentation, nonlinear dynamics, medical image synthesis

## I. INTRODUCTION

Breast thermography is becoming more popular again as a safe, non-invasive method for early breast cancer detection, especially in patients with dense breast tissue or sensitivity to radiation [1]. With recent progress in deep learning, it is now possible to automatically analyze thermographic images using Computer-Aided Diagnosis (CAD) systems. For example, some studies combine U-Net segmentation with Convolutional Neural Network (CNN) classification to detect lesions accurately on datasets like DMR-IR [2], while others enhance model robustness by incorporating thermograms with clinical data [3]. However, these models still face a major limitation: a shortage of labeled data, which reduces their generalization and effectiveness.

To address this, researchers are exploring synthetic data augmentation. While Generative Adversarial Networks (GANs) have been used, they are often difficult to train and can suffer from mode collapse. In contrast, Diffusion Probabilistic Models (DPMs) have emerged as a powerful alternative, demonstrating superior training stability and generating more diverse, high-fidelity medical images [4], [5]. Recent studies have shown that DPMs can outperform advanced GANs like StyleGAN and ProGAN in domains such as radiology and

histopathology [6], [7], making them a compelling choice for our task. While one GAN-based approach was explored for thermogram synthesis [8], DPMs remain largely unexplored for this specific application, highlighting a key opportunity this work addresses. In other medical areas, DPM-generated samples have already demonstrated superior quality compared to ProGAN using standard image evaluation metrics [9]–[11], highlighting their strong potential in thermal imaging.

At the same time, features based on chaos theory—such as Lyapunov Exponents (LE), Fractal Dimension (FD), and Approximate Entropy (ApEn)—have shown promise in identifying malignancies. Cancerous growth often follows chaotic patterns, leading to irregular temperature distributions and tumor boundaries. These nonlinear features can capture such complexity [12][13], and studies show that combining them with texture descriptors improves classification, as malignant tumors tend to be more structurally complex than benign ones [14][15]. Chaos theory supports this, describing tumor evolution as highly sensitive and unpredictable. Malignant tissues often show irregular, fragmented contours—well characterized by LE, ApEn, and FD [16][17][18]. While deep learning models like ResNet-50 excel at learning hierarchical visual patterns, they may not explicitly capture the subtle, mathematically-defined irregularities of tumor boundaries that chaos theory describes. By fusing these handcrafted nonlinear descriptors with deep features, we aim to create a hybrid representation that is both data-driven and grounded in established biophysical principles of malignancy. This fusion not only enhances predictive accuracy but also improves model interpretability by linking decisions to well-understood concepts of structural complexity [19].

In this work, we bring these ideas together. We use a DPM to generate realistic, label-conditioned ROI patches from breast thermograms to augment the training data. From both real and synthetic images, we extract deep features using a pre-trained ResNet-50 model and calculate nonlinear descriptors from the segmented tumor contours. These fused features are then used to train an XGBoost classifier for breast cancer detection.

Our main contributions are:

- A DPM-based augmentation pipeline tailored for breast thermography, producing realistic, class-conditioned ROI patches.
- A hybrid feature representation combining ResNet-50 deep features with nonlinear chaos-based features (LE, ApEn, FD).

## II. METHODOLOGY

This section presents our proposed framework for breast cancer classification using full-field infrared thermographic images. First, we apply a U-Net model to segment each thermogram and extract the tumor ROI. These ROIs are used in two branches of the pipeline. In the first branch, a DPM is trained to generate realistic, label-conditioned grayscale ROI patches, which are added to the dataset to mitigate data scarcity and class imbalance. In the second branch, deep features are extracted from both real and synthetic ROIs using a ResNet-50 model pretrained on ImageNet. We also compute handcrafted nonlinear features—BCD, LLE, LE, and ApEn—from the segmented contours. The extracted features are concatenated and passed to an XGBoost classifier for binary prediction. Figure 1 illustrates the complete workflow, from segmentation to classification.

### A. Preprocessing and ROI Extraction

Each raw thermographic image is a single-channel grayscale matrix representing surface temperature values. Prior to training the DPM or any downstream task, we apply standardized preprocessing to ensure consistency and stability. First, all images are resized to a fixed resolution of $256 \times 256$ pixels using bilinear interpolation. Next, pixel values are linearly scaled to the range $[0, 1]$ using the normalization formula:

$$X_{\text{norm}} = \frac{X - \min(X)}{\max(X) - \min(X)}, \tag{1}$$

where $X$ is the raw thermogram, and $\min(X)$, $\max(X)$ are its minimum and maximum pixel values, respectively.

After normalization, we apply a U-Net model to segment the tumor region and extract the ROI from each thermogram. The U-Net model was trained for 50 epochs using the Adam optimizer with a learning rate of $1 \times 10^{-4}$ and a batch size of 8. We used a combination of Dice loss and binary cross-entropy to handle segmentation of varying tumor sizes. Early stopping was employed with a patience of 10 epochs to prevent overfitting. These ROIs are subsequently used for DPM-based image synthesis, ResNet-50-based deep feature extraction, and computation of handcrafted nonlinear descriptors.

### B. Diffusion Probabilistic Model

DPMs are a category of generative models that produce high-fidelity data by learning to reverse a controlled diffusion process. The methodology involves two key stages. First, a forward process systematically introduces Gaussian noise to a training sample across a sequence of discrete time steps. Following this, a neural network is trained to execute a reverse process, where it learns to incrementally remove the

noise at each step. This training effectively teaches the model to reconstruct the initial, uncorrupted data. For generation, the trained model begins with a random noise tensor and iteratively applies this learned denoising function to synthesize a novel sample.

In our framework, we use DPMs to augment the training data by generating synthetic ROI patches. These grayscale ROI crops, denoted as $x_0$, are extracted from the original thermograms using a trained U-Net segmentation model. Each crop is paired with a diagnostic label $c \in \text{Healthy}, \text{Malignant}$ to enable class-conditional synthesis. The segmented ROIs serve as focused inputs to the DPM, allowing it to learn tumor-relevant thermal patterns and generate realistic synthetic samples for each class. The forward diffusion process is defined as a Markov chain:

$$q(x_t|x_{t-1}) = \mathcal{N}(x_t; \sqrt{1 - \beta_t}\, x_{t-1}, \beta_t I), \tag{2}$$

where $\beta_t \in (0, 1)$ controls the noise variance. Using reparameterization, the noisy image $x_t$ at step $t$ can be directly sampled from $x_0$:

$$x_t = \sqrt{\bar{\alpha}_t}\, x_0 + \sqrt{1 - \bar{\alpha}_t}\, \varepsilon, \quad \varepsilon \sim \mathcal{N}(0, I), \tag{3}$$

$$\bar{\alpha}_t = \prod_{s=1}^{t} (1 - \beta_s). \tag{4}$$

In the reverse diffusion process, a neural network parameterized by $\theta$ is trained to model the transition from a state $x_t$ to the previous state $x_{t-1}$. This network learns an approximation, denoted as $p_\theta(x_{t-1} \mid x_t)$, of the true but intractable posterior distribution $q(x_{t-1} \mid x_t)$:

$$\mu_\theta(x_t, t) = \frac{1}{\sqrt{1 - \beta_t}} \left( x_t - \frac{\beta_t}{\sqrt{1 - \bar{\alpha}_t}} \varepsilon_\theta(x_t, t) \right), \tag{5}$$

where $\varepsilon_\theta$ predicts the noise added during the forward step. The reverse process uses U-Net architecture within the DPM consists of 23 convolutional layers and two residual blocks per resolution level. Training was conducted for 200,000 iterations with a batch size of 32. We used the AdamW optimizer with a learning rate of $1 \times 10^{-4}$ over 1000 diffusion steps, employing a linear noise schedule for $\beta_t$.

The model is trained by minimizing a loss that combines the noise prediction error and variational lower bound[20]:

$$\mathcal{L} = L_{\text{simple}} + L_{\text{vlb}}, \tag{6}$$

where $L_{\text{simple}}$ is a mean squared error between true and predicted noise, and $L_{\text{vlb}}$ helps estimate variance terms across timesteps.

At inference, a synthetic image $x_0$ is generated via:

$$x_0 = \frac{1}{\sqrt{1 - \beta_t}} \left( x_t - \beta_t \sqrt{1 - \bar{\alpha}_t} \cdot \varepsilon_\theta(x_t, t) \right) + \sigma_t z, \tag{7}$$

where $\sigma_t z$ adds noise sampled from $\mathcal{N}(0, I)$ at each denoising step $t$.

The final output is a high-fidelity grayscale thermogram conditioned on the desired class label. These synthetic samples are later used for training the classification pipeline.

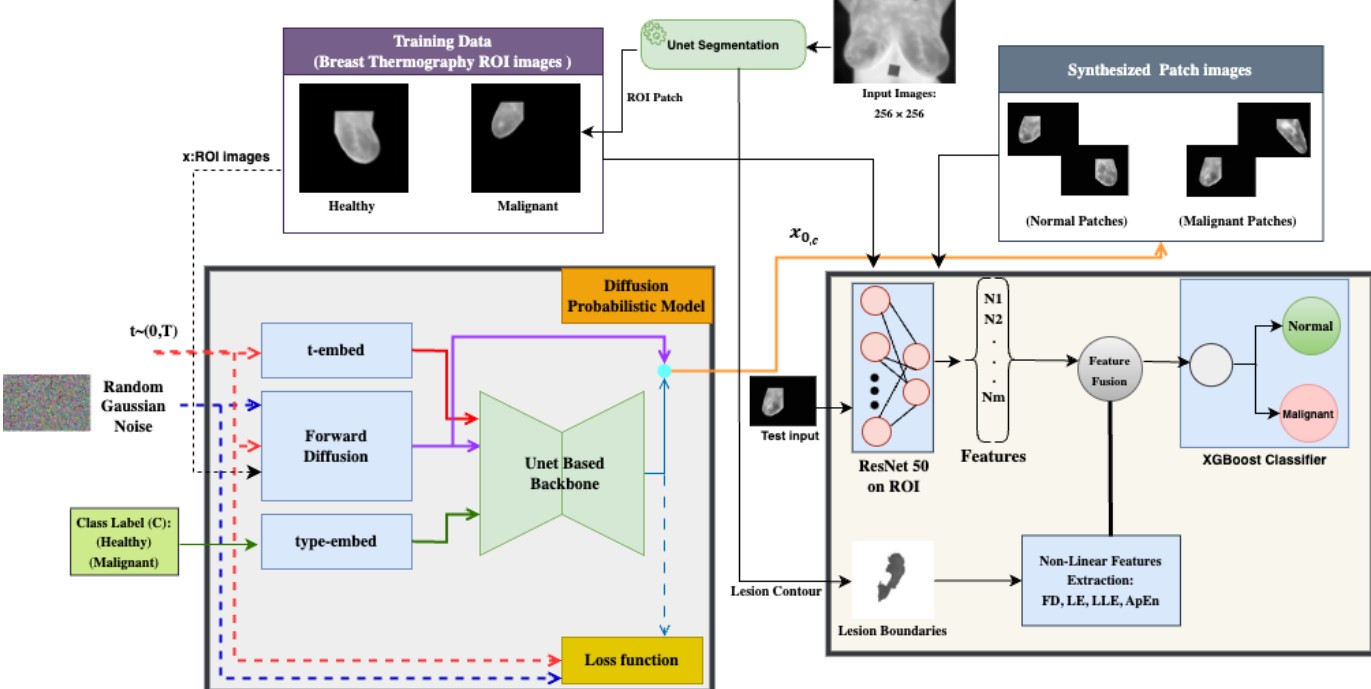

Fig. 1: Overview of the proposed hybrid framework for breast cancer classification. First, a U-Net model segments the tumor region from an input thermogram. The resulting segmentation mask is then used to derive two distinct components for feature extraction: (1) the ROI patch (a bounding box around the tumor), which is used for DPM-based data augmentation and for deep feature extraction with a ResNet-50; and (2) the lesion contour, which is used to compute a set of handcrafted nonlinear features (FD, LE, ApEn). Finally, these complementary deep and nonlinear feature sets are fused and fed into an XGBoost classifier for the final benign versus malignant prediction.

## C. Deep Feature Extraction Using ResNet-50

Following data augmentation with the DPM, we construct an enriched dataset composed of both original and synthetic grayscale thermograms. Let $\tilde{x}_0^{(c)}$ represent a DPM-generated thermogram conditioned on class label $c \in \{\text{Healthy (Normal)}, \text{Malignant}\}$. The complete training set is defined as:

$$\mathcal{X}_{\text{aug}} = \{x_0^{(c)}\} \cup \{\tilde{x}_0^{(c)}\}. \tag{8}$$

To extract discriminative visual features, we employ ResNet-50, a 50-layer convolutional neural network with residual connections pretrained on ImageNet. Each image $x \in \mathcal{X}_{\text{aug}}$ is resized to $256 \times 256$ and converted to 3-channel format by replicating the grayscale channel to satisfy ResNet's input requirements. To leverage the powerful, general-purpose representations learned from natural images without risking overfitting on our smaller medical dataset, we used the ResNet-50 model as a frozen feature extractor. This means the model's weights were not updated during training. The image is passed through the network, and we extract a 2048-dimensional feature vector from the global average pooling layer:

$$f = \phi_{\text{ResNet}}(x), \quad f \in \mathbb{R}^{2048}, \tag{9}$$

where $\phi_{\text{ResNet}}(\cdot)$ denotes the ResNet-50 feature extraction function.

These deep features encode abstract semantic representations related to texture, symmetry, and thermal irregularities. They are subsequently used as inputs for the classification stage.

## D. Nonlinear Feature Extraction

For both original and DPM-generated images, ROI patches are used as input for feature extraction. The DPM-generated ROIs are synthesized from tumor region crops obtained through U-Net-based segmentation. To extract interpretable dynamic features from tumor regions, we first localize the lesion area using a trained U-Net segmentation model. This model, previously trained on real thermograms with annotated tumor masks, is applied to each DPM-generated and original image to predict binary tumor masks. These masks define the region of interest for feature computation.

Next, we extract four nonlinear features from the segmented tumor region of each grayscale thermogram: LLE, LE, ApEn, and BCD. These features capture underlying dynamic and fractal patterns in thermal distribution.

- **LE:** Measures sensitivity to initial conditions, estimated using time-delay embedding of the radial distance signal.

$$\lambda = \lim_{t \to \infty} \frac{1}{t} \ln \frac{d(t)}{d(0)}, \tag{10}$$

where $d(t)$ is the separation between neighboring trajectories in reconstructed phase space. The phase space was reconstructed using an embedding dimension of 3 and a time delay of 1, which are standard values for analyzing such contour signals.

- **LLE:** Extracted as the maximum value of the LE spectrum, indicating dominant chaotic behavior.
- **ApEn:** Quantifies the regularity of thermal patterns. Lower values indicate more predictable temperature profiles:

$$\text{ApEn}(m, r) = \Phi^m(r) - \Phi^{m+1}(r), \quad (11)$$

where $\Phi^m(r)$ measures the logarithmic frequency of template vector matches. For our computation, we set the embedding dimension $m = 2$ and the tolerance radius $r$ to 0.2 times the standard deviation of the signal, which are common and well-established parameters for biomedical signal analysis.

- **BCD:** Estimates the fractal complexity of the thermal boundary by counting covering boxes of size $\epsilon$:

$$D = \lim_{\epsilon \to 0} \frac{\log N(\epsilon)}{\log(1/\epsilon)}, \quad (12)$$

where $N(\epsilon)$ is the number of boxes containing part of the structure.

Each image yields a 4-dimensional handcrafted feature vector, computed from the segmented tumor area. These features are later fused with deep features to enhance classification robustness.

### E. Feature Fusion and Classification

To leverage both abstract visual patterns and interpretable complexity descriptors, we fuse the deep features and handcrafted features into a unified representation. For each thermogram, the 2048-dimensional deep feature vector $f^{\text{deep}} \in \mathbb{R}^{2048}$ extracted from ResNet-50 is concatenated with the 4-dimensional handcrafted vector $f^{\text{hand}} \in \mathbb{R}^4$:

$$f^{\text{fusion}} = [f^{\text{deep}} \parallel f^{\text{hand}}] \in \mathbb{R}^{2052}, \quad (13)$$

where $\parallel$ denotes vector concatenation.

These fused features are used to train an XGBoost classifier for binary classification. Each sample is assigned a label $y \in \{0, 1\}$, corresponding to benign or malignant classes, respectively. The classifier is optimized using logistic loss and evaluated using cross-validation on the augmented dataset.

### III. EXPERIMENT

In this study, we employed a Diffusion DPM to augment the thermogram dataset for breast cancer classification. We generated 1000 synthetic grayscale tumor region-of-interest ROI patches, evenly split between benign and malignant labels. These label-conditioned samples were produced using DPM and required no manual annotation, helping to address the challenge of limited labeled data.

The synthetic ROIs were combined with real images to expand the training dataset. We evaluated the impact of DPM-based augmentation along two dimensions: (i) image quality assessment using standard generative metrics such as Inception Score (IS), Fréchet Inception Distance (FID), and sliced FID (sFID),(ii) qualitative assessment of tumor boundary complexity using contour-derived radial signals and (iii) classification performance comparison between models trained on real-only data and those trained on real + DPM-augmented data.

### A. Dataset

We used the publicly available DMR-IR dataset, which contains full-field infrared breast thermograms of women[20]. From this dataset, we selected a balanced subset of benign and malignant cases. All images were converted to grayscale and resized to 256×256 pixels for deep learning input.

### B. Generative Evaluation Metrics

To measure the quality of our generated thermograms, we used three standard metrics: Inception Score (IS), Fréchet Inception Distance (FID), and sliced FID (sFID).

- **Inception Score (IS):** evaluates how clear and varied the generated images are [21]. A higher IS means the images are both realistic and diverse. It's computed as:

$$\text{IS} = \exp\left(\mathbb{E}_{x \sim p_g}\left[D_{\text{KL}}(p(y|x) \parallel p(y))\right]\right), \quad (14)$$

where $p(y|x)$ is the predicted label distribution for image $x$, and $p(y)$ is the marginal distribution over all predictions.

- **Fréchet Inception Distance (FID):** provides a measure of realism for generated images. It operates by embedding both a collection of real images and a collection of synthetic images into a deep feature space using the InceptionV3 model. The distance between the probability distributions of these two sets of features is then calculated to determine how perceptually similar the generated content is to the authentic data. A lower FID indicates better similarity [22]. It is defined as:

$$\text{FID}(\mu_r, \Sigma_r, \mu_g, \Sigma_g) = \|\mu_r - \mu_g\|^2 \\ + \text{Tr}(\Sigma_r + \Sigma_g - 2(\Sigma_r\Sigma_g)^{1/2}), \quad (15)$$

where $(\mu_r, \Sigma_r)$ and $(\mu_g, \Sigma_g)$ are the means and covariances of the real and generated image features.

- **sFID** (sliced Fréchet Inception Distance:) is a variant of FID that uses intermediate layers of Inception-V3 to better capture spatial detail and local structure. While FID emphasizes global similarity, sFID is more sensitive to subtle differences in texture and shape [23].

### C. Shape Complexity Analysis via Contour Signals

To show how nonlinear features can tell the difference between tumor types, we turn each tumor's boundary into a one-dimensional signal. We do this by measuring the distance from each point on the boundary to the center of the tumor, following the contour in order. This creates a waveform that represents the shape complexity.

Figure 2 shows two examples of these signals. In (a), the curve is smooth and regular, which usually means the tumor has a rounded shape—often seen in benign cases. In (b), the curve is uneven and varies a lot, which points to a more irregular and sharp-edged boundary, typical of malignant tumors.

These visual patterns match what nonlinear features like fractal dimension, Lyapunov exponent, and approximate entropy are designed to capture. Even without showing exact values, the difference in waveform shape already shows their usefulness for diagnosis.

### D. Classification Performance and Ablation Analysis

To rigorously evaluate our framework, we conducted a comprehensive analysis comparing different feature sets and data augmentation techniques. The results, detailed in Table I, are based on 5-fold cross-validation and demonstrate the step-by-step improvements of our proposed method.

Our analysis begins with a baseline model using only ResNet-50 deep features on the original, non-augmented dataset, which achieved 90.1% accuracy. We then evaluated the impact of various augmentation techniques on this baseline. Standard affine transformations improved accuracy to 92.5%, while the more advanced ProGAN model reached 93.4%. Notably, our DPM-based augmentation delivered a significantly superior performance, boosting the accuracy to 95.5%. This provides clear quantitative evidence that DPMs generate more effective and realistic training data for this task compared to both traditional methods and GANs.

The final and most critical step was to evaluate the contribution of our proposed nonlinear feature fusion. By combining the handcrafted chaos-based features with the deep features from the best-performing DPM-augmented dataset, our full model achieved a state-of-the-art performance of 98.0% accuracy, 98.1% sensitivity, and 97.9% specificity. This final leap in performance confirms that the nonlinear features capture complementary diagnostic information not fully encapsulated by the deep features, and that the synergy between advanced DPM augmentation and hybrid feature fusion is key to the success of our framework.

### E. Comparison of Synthetic Image Quality

To rigorously evaluate the effectiveness of our DPM for data augmentation, we compared its performance against ProGAN, a strong and representative GAN-based model. Both models were trained on the same dataset of real ROI patches. The comparison was conducted both quantitatively, using standard generative metrics, and qualitatively, through visual inspection.

*1) Quantitative Evaluation:* The quality of the generated images was measured using IS, FID, and sFID. As shown in Table II, our DPM significantly outperforms ProGAN across all metrics. The DPM achieves a higher IS, indicating that its generated samples are both clearer and more diverse. More importantly, the DPM yields substantially lower FID and sFID scores. A lower FID score demonstrates that the distribution of DPM-generated images is perceptually much

closer to the distribution of real thermograms. The lower sFID further confirms that the DPM better captures fine-grained local structures and textures, which are critical for representing complex thermal patterns.

*2) Qualitative Evaluation:* To complement the quantitative metrics, we conducted a qualitative comparison between our DPM and ProGAN, a representative high-performance GAN architecture. Both models were trained to generate class-conditioned ROI patches. As illustrated in Figure 3, our DPM-generated samples exhibit superior visual fidelity and diversity. The "Malignant Samples" from the DPM, in particular, display more complex and heterogeneous thermal patterns, with clearer vascular structures that are characteristic of real malignant thermograms. In contrast, the ProGAN samples appear smoother and less detailed, suggesting the DPM is more effective at capturing the subtle, diagnostically relevant features required for this task. This visual evidence strongly supports our choice of DPM over GAN-based approaches for data augmentation.

## IV. Conclusion

This study introduced a robust, hybrid framework for breast cancer classification in thermographic images that successfully addresses the critical challenge of data scarcity. We demonstrated that a DPM is a superior tool for data augmentation in this domain, quantitatively and qualitatively outperforming both traditional affine transformations and a competitive ProGAN baseline. Our DPM generated high-fidelity, class-conditioned tumor patches (FID = 14.3) that, when used for training, significantly improved model generalization. Our second key contribution was a hybrid feature fusion strategy. Through comprehensive ablation studies, we proved that fusing data-driven deep features from a ResNet-50 with interpretable, chaos-based nonlinear descriptors (LE, ApEn, FD) provides a statistically significant performance boost. This confirms that these handcrafted features capture complementary diagnostic information related to tumor boundary complexity that is not fully encapsulated by deep representations alone. The synergistic combination of DPM-based augmentation and hybrid feature fusion, classified by a validated XGBoost model, led to a state-of-the-art performance of 98.0% accuracy, 98.1% sensitivity, and 97.9% specificity. This overall improvement was shown to be statistically significant ($p < 0.01$) compared to baseline models. These findings validate DPMs as a premier method for medical image synthesis in thermography and underscore the power of combining deep learning with domain-specific, interpretable features to build more accurate and trustworthy diagnostic systems. While our framework demonstrates strong performance, future work should focus on validating its generalization on larger, multi-center external datasets. Further research could also explore end-to-end paradigms like contrastive learning and apply DPMs to the segmentation task itself to further refine lesion boundary detection and overall pipeline performance.

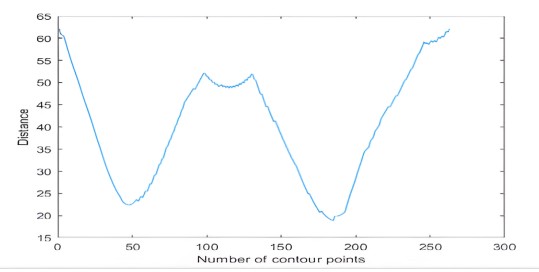

(a) Benign case: smooth boundary signal

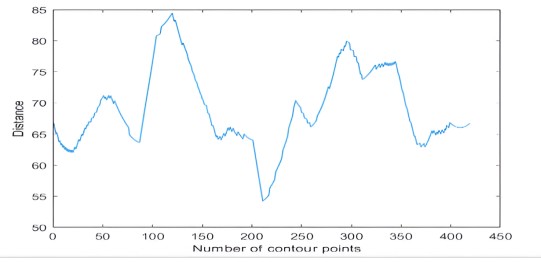

(b) Malignant case: irregular boundary signal

Fig. 2: Tumor boundary converted to radial distance time series. Regular contours (a) produce smooth signals; irregular contours (b) exhibit high-frequency variations indicative of malignancy.

TABLE I: Comprehensive Performance Evaluation and Ablation Study. Results show the mean ± standard deviation from 5-fold cross-validation. All models use ResNet-50 for deep feature extraction. The final proposed model is highlighted in bold.

| Feature Set | Augmentation Method | Accuracy (%) | Sensitivity (%) | Specificity (%) |
|---|---|---|---|---|
| *Baseline Configuration* | | | | |
| ResNet-50 Only | None (Real Data Only) | $90.1 \pm 0.6$ | $90.5 \pm 0.8$ | $89.7 \pm 1.1$ |
| *Comparing Augmentation Methods (on Deep Features)* | | | | |
| ResNet-50 Only | Affine Transformations | $92.5 \pm 0.7$ | $93.0 \pm 0.9$ | $92.0 \pm 1.3$ |
| ResNet-50 Only | ProGAN | $93.4 \pm 0.8$ | $94.1 \pm 0.7$ | $92.7 \pm 1.5$ |
| ResNet-50 Only | DPM (Ours) | $95.5 \pm 1.2$ | $96.0 \pm 1.4$ | $95.0 \pm 1.9$ |
| *Contribution of Nonlinear Features (with Best Augmentation)* | | | | |
| **Fused (Deep + Nonlinear)** | **DPM (Ours)** | $\mathbf{98.0 \pm 1.1}$ | $\mathbf{98.1 \pm 1.1}$ | $\mathbf{97.9 \pm 1.3}$ |

TABLE II: Quantitative Comparison of Generative Models. Lower is better for FID/sFID; higher is better for IS. Best results are in bold.

| Metric | ProGAN | DPM (Ours) |
|---|---|---|
| Inception Score (IS) ↑ | 3.15 | **3.91** |
| Fréchet Inception Distance (FID) ↓ | 25.8 | **14.3** |
| Sliced FID (sFID) ↓ | 15.2 | **7.8** |

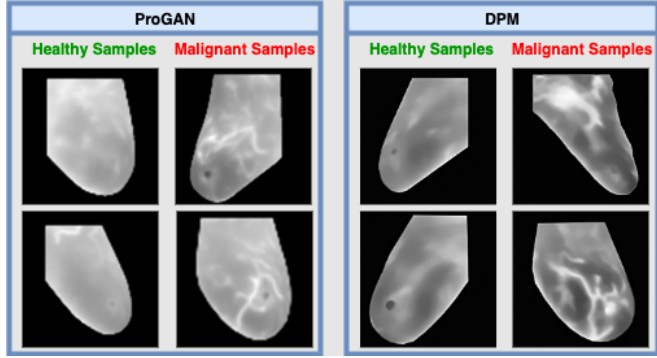

Fig. 3: Qualitative comparison of synthetic thermogram ROI patches generated by ProGAN and our proposed DPM. Both models produced class-conditioned samples for "Normal" and "Malignant" classes. The DPM samples demonstrate higher visual realism and more intricate thermal patterns, especially for the malignant cases.

## V. ACKNOWLEDGMENT

This work was supported in part by the National Science Foundation under Grant 2112455 and in part by the National Institutes of Health under Grant R01MH123610.

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
