# OpenReview forum: "Breast Cancer Detection in Thermographic Images via Diffusion‐Based Augmentation and Nonlinear Feature Fusion"
_IEEE.org/EMBS/BHI/2025/Conference — BHI 2025_

### Official Review · Reviewer_avLZ · 2025-07-17
**The paper, diffusion‑augmented thermographic breast‑cancer classification, shows strong empirical gains but lacks baseline comparisons and statistical testing**

**Confidence:** 5
**Clarity Of Writing:** good
**Clinical Significance:** fair
**Methodological Novelty:** good
**Overall Rating:** 5
**Final Rating:** 6

**Experiments And Results:**

fair

**Questions For The Authors:**

- Can you provide paired tests (e.g., McNemar, bootstrap CI) to support the reported 7% gain?

- Did you replicate the BCD-GAN from Veerlapalli & Dutta (2025), or compare against any other conventional augmentation methods (e.g., GANs)?

- Have you conducted an ablation study to quantify the added value of the chaos-based features over using deep features alone?

**Strengths:**

- Novel use of DPMs in thermographic augmentation, where GANs were typically used

- Large performance gains, with +7% accuracy/sensitivity using DPM data

**Summary Of The Paper:**

The paper segments tumor ROIs from DMR-IR thermograms using U-Net, generates 1,000 synthetic ROI patches via a 1,000-step DPM, and combines them with real data. It extracts 2048-dim ResNet-50 deep features and fuses them with 4 handcrafted chaos features (LE, LLE, ApEn, BCD). An XGBoost classifier trained on this fused feature set achieves 98% accuracy and sensitivity (vs. 91% baseline). Generative quality is validated using IS (3.91), FID (14.3), and sFID (7.8).

**Weaknesses:**

- No statistical significance testing is reported to verify that the 7% gain is not due to chance

- Missing comparison to GAN-based or traditional augmentations, despite claiming DPM superiority

- Lack of hyperparameter justification (e.g., 1,000 diffusion steps, LR = 1e‑4), weakening reproducibility

---

### Official Review · Reviewer_kpcB · 2025-07-17
**A Promising Approach with Minor Revisions Needed**

**Confidence:** 3
**Clarity Of Writing:** great
**Clinical Significance:** great
**Methodological Novelty:** great
**Overall Rating:** 7

**Experiments And Results:**

good

**Questions For The Authors:**

see my comments in Weaknesses

**Strengths:**

- This paper addresses the issue of data scarcity in medical imaging, particularly in breast cancer thermography, through synthetic data generation.
- The use of DPMs for data augmentation has led to a significant improvement in classification performance.
- This approach achieves promising results while using relatively simple and straightforward model architectures.
- In addition to the classification metrics, the paper reports quality metrics for the generated samples.

**Summary Of The Paper:**

In this paper, diffusion model-based data augmentation combined with a set of nonlinear features is used to improve classification performance for breast cancer detection using thermographic images. A U-Net model is employed to generate tumor region patches along with their corresponding labels. Features extracted from a ResNet-50 model applied to both real and diffusion-generated images, are fused with nonlinear descriptors to enhance classification accuracy. The results show improved performance compared to the non-augmented approach.

**Weaknesses:**

- Although DPMs are known to outperform GANs in image generation tasks, including medical imaging, adding a comparison of metrics on the overall performance with GAN and DPM generated data would highlight the advantage of using DPM over GAN.
- An ablation study including the individual contribution of ResNet features and nonlinear features, both with and without augmentation can be included to show how each component contributes to the overall performance.
- Although GAN and DPM are well-known abbreviations, the authors may consider expanding them when first mentioned in the introduction to help readers who may not be familiar with these terms.

---

### Official Review · Reviewer_PSi1 · 2025-07-17
**Breast Cancer Detection in Thermographic Images via Diffusion-Based Augmentation and Nonlinear Feature Fusion**

**Confidence:** 3
**Clarity Of Writing:** good
**Clinical Significance:** good
**Methodological Novelty:** fair
**Overall Rating:** 5
**Final Rating:** 6

**Experiments And Results:**

fair

**Questions For The Authors:**

N/A

**Strengths:**

1. Novel use of DPM in thermal medical imaging: The application of a diffusion probabilistic model for augmenting thermographic breast cancer images is novel and well-motivated, offering a promising alternative to GAN-based synthesis approaches.
2. Strong performance: The proposed method achieves high classification metrics (98% accuracy, 98% sensitivity), demonstrating its potential utility in real-world diagnostic applications.
3. Well-structured pipeline: The workflow is clearly defined, from segmentation to augmentation, feature extraction, and classification, facilitating reproducibility.

**Summary Of The Paper:**

This paper presents a breast cancer classification framework based on grayscale thermographic images. To address the challenge of limited annotated data, the authors adopt a diffusion probabilistic model (DPM) to generate class-conditioned synthetic tumor region-of-interest (ROI) patches. These synthetic images are combined with real thermograms for training. The classification pipeline extracts deep features using a pretrained ResNet-50 model and combines them with nonlinear handcrafted features (e.g., Lyapunov Exponent, Fractal Dimension, Approximate Entropy) to capture structural complexity. An XGBoost classifier is used for final prediction. Experimental results on the DMR-IR dataset show that the proposed method significantly improves classification accuracy and sensitivity.

**Weaknesses:**

1. Lack of Baseline Comparisons: The paper does not benchmark the proposed diffusion probabilistic model (DPM) against other widely used data augmentation techniques, such as GAN-based models (e.g., StyleGAN, ProGAN) or traditional approaches (e.g., affine transformations, noise injection). As a result, the relative advantage of DPM remains unclear.
2. Unjustified Classifier Choice: The paper employs XGBoost for classification without explaining the rationale behind this choice. It remains unclear why XGBoost is preferred over other commonly used machine learning models (e.g., SVM, random forest, shallow neural networks), and whether this choice has a significant impact on performance.  It would be better to have a comparison experiment.
3. Limited Generalization Evaluation: All experiments are conducted solely on the DMR-IR dataset. The absence of cross-dataset or external validation limits confidence in the method’s generalizability to other clinical settings or datasets.
4. Poor Figure Quality: Figure 2 is poorly rendered—panel (b) appears distorted, and its x-axis label is partially obscured. This affects the clarity and interpretability of the presented shape complexity analysis.

---

### Official Review · Reviewer_LHoQ · 2025-07-18
**Diffusion based sample synthesis as augmentation for tumor classification**

**Confidence:** 4
**Clarity Of Writing:** good
**Clinical Significance:** good
**Methodological Novelty:** good
**Overall Rating:** 5

**Experiments And Results:**

fair

**Questions For The Authors:**

- Why manual feature extraction instead of simply using a Resnet? If the hypothesis hinges entirely on the fractal dimension and so on, that is at least partly a function of the annotation style and consistency in the dataset the segmentation UNet was trained on since the decision boundary relies on the differences in irregularity of the radial distance signal and human annotation is stochastic.
- Why not formulate it as multiclass segmentation instead of segmentation followed by classification?
- The closest architectural analog that comes to mind is one formulated as a contrastive learning problem. Was this considered? Contrastive learning would yield the features that maximize discriminative ability for the problem and distribution baking in the least inductive biases. I would recommend something like:

| Approach | 1 | 2 | 3 | 4 | 5 | 6 | 7 | 8 |
|----------|---|---|---|---|---|---|---|---|
| **Segmentation** |
| Binary U-Net | ✓ | ✓ | ✓ | ✓ | ✗ | ✗ | ✗ | ✗ |
| Multiclass U-Net (Benign/Malignant/Background) | ✗ | ✗ | ✗ | ✗ | ✓ | ✓ | ✗ | ✗ |
| **Learning Method** |
| Supervised (ImageNet ResNet) | ✓ | ✓ | ✗ | ✗ | ✗ | ✗ | ✗ | ✗ |
| Contrastive Learning | ✗ | ✗ | ✓ | ✓ | ✗ | ✗ | ✓ | ✓ |
| **Feature Extraction** |
| Deep Features | ✓ | ✓ | ✓ | ✓ | N/A | N/A | ✓ | ✓ |
| Nonlinear Features | ✓ | ✗ | ✗ | ✓ | N/A | N/A | ✗ | ✗ |
| **Augmentation** |
| DPM | ✓ | ✗ | ✗ | ✓ | ✗ | ✓ | ✗ | ✓ |
| **Classification** |
| XGBoost | ✓ | ✓ | ✓ | ✓ | N/A | N/A | ✓ | ✓ |
| Direct from Segmentation | ✗ | ✗ | ✗ | ✗ | ✓ | ✓ | ✗ | ✗ |
| **End-to-End** | ✗ | ✗ | ✗ | ✗ | ✓ | ✓ | ✓ | ✓ |

**Strengths:**

- Well-motivated
- The paper demonstrates familiarity with the State of the Art of image generation and effectively leverages domain specific insights
- Language is clear

**Summary Of The Paper:**

The paper proposes a Denoising Diffusion Probabilistic Model based method to augment the dataset for a two-step classification pipeline consisting of:
1. an initial UNet to extract tumor boundary given a thermographic image. ROI bounding boxes are computed based on the segmentation masks of masses.
2. An XGBoost model trained to classify masses extracted from thermographic scans of breast cancer patients.

Once tumor boundaries are segmented manual feature extraction is performed, motivated by the established difference in entropy between the radial distance function for benign and malignant masses. These manually extracted features are fused with features extracted using a pretrained ResNet and then classified.

The DDPM is used to synthesize ROI patches from which tumor boundary extraction is performed and DL-features and nonlinear features are extracted and fused, drastically reducing the amount of annotation required.

**Weaknesses:**

- Figure color palette could be less jarring, figures could use annotations for dimensions at every stage of the pipeline
- “to the original thermograms to ROI patches” - Fig 1. caption - missing verb
- Separate forward and reverse process diagrams for DDPM use and how they fit into the full pipeline would make the work more accessible
- I would recommend tweaking the figure so the relationship between tumor boundaries and the ROIs they belong to is clearer, specifically which is derived from which and using consistent representations for the UNet used in the DDPM and for segmentation.
- The overall method, formulation and motivation could be a bit clearer, particularly, the stages at which each type of prediction - segmentation mask, scalar label etc. are made.
- An ablation study to compare, multiclass UNet that combines the segmentation and classification tasks into a single forward pass, DL feature extraction only, fused features etc, would be desirable to validate the explanation of observed improvement in model performance.
- Unknowns without ablation
    - Whether the nonlinear chaos-based features actually improve performance
    - If ResNet-50 alone would achieve similar 98% accuracy
    - The relative importance of deep vs. handcrafted features

---

### Official Review · Reviewer_Zjh7 · 2025-07-18
**The paper presents a conceptually strong and innovative framework. However, a lack of methodological detail and the details of evaluation should be fixed before publication.**

**Confidence:** 4
**Clarity Of Writing:** good
**Clinical Significance:** great
**Methodological Novelty:** good
**Overall Rating:** 4
**Final Rating:** 6

**Experiments And Results:**

poor

**Questions For The Authors:**

Was the pre-trained ResNet-50 model fine-tuned on your thermographic dataset, or was it used purely as a frozen feature extractor?
The paper reports the IS, FID, and sFID for the diffusion-generated thermograms. What would be considered good values in this specific domain?
Could the authors specify the exact parameter settings used for the nonlinear feature extraction methods?

**Strengths:**

The paper tackles the significant challenge of limited labeled data in medical imaging, specifically for breast thermography, which is an important bottleneck for deep learning adoption in this field.

The fusion of deep features (from ResNet-50) with interpretable nonlinear, chaos-based features (Lyapunov Exponents, Approximate Entropy, Fractal Dimension) is a compelling strategy.

Focusing on thermography (a non-invasive, radiation-free method) offers a patient-friendly alternative for breast cancer screening.

**Summary Of The Paper:**

This paper presents a framework for breast cancer detection in infrared thermographic images using a two-part approach. First, it addresses data scarcity by generating realistic, label-conditioned synthetic tumor ROI patches with a Diffusion Probabilistic Model (DPM), trained on U-Net-segmented regions. Second, it combines deep features from a ResNet-50 with nonlinear descriptors, such as Fractal Dimension, Lyapunov Exponents, and Approximate Entropy, extracted from tumor boundaries. These fused features are input to an XGBoost classifier. The system achieves 98% accuracy using cross-validation, highlighting the effectiveness of DPM-based augmentation and hybrid feature fusion for explainable thermal imaging-based cancer detection.

**Weaknesses:**

The paper does not provide hyperparameters for DPM, U-Net, XGBoost, or nonlinear feature extraction (e.g., ApEn parameters). These should be mentioned for reproducibility.

If the DPM was trained on the entire real dataset (or even folds that include data that would later be in a test fold) to generate synthetic images, and these synthetic images are then added to the training folds for cross-validation, the model could indirectly "see" information from the test set. The authors should add details on how the DMR-IR dataset was split into training, validation, and test sets. My main concern is this part as accuracy of 98% seems a bit high for thermography, which is known to be challenging due to inter-subject variability.

While "Real Only" is a good internal baseline, the paper could benefit from a more direct quantitative comparison against other published methods on the DMR-IR dataset that don't use DPMs or hybrid features, to further demonstrate the incremental value of the proposed approach.

The authors report IS, FID, and sFID values for the DPM-generated thermograms; however, they should also provide comparative references or benchmarks from thermographic or related medical imaging domains to contextualize whether the acheived values reflect strong generative performance.